# Enabling Security Services in Socially Assistive Robot Scenarios for Healthcare Applications

**DOI:** 10.3390/s21206912

**Published:** 2021-10-18

**Authors:** Alexandru Vulpe, Răzvan Crăciunescu, Ana-Maria Drăgulinescu, Sofoklis Kyriazakos, Ali Paikan, Pouyan Ziafati

**Affiliations:** 1Telecommunications Department, University Politehnica of Bucharest, 061071 Bucharest, Romania; alex.vulpe@radio.pub.ro (A.V.); razvan.craciunescu@upb.ro (R.C.); ana.dragulinescu@upb.ro (A.-M.D.); 2Innovation Sprint Sprl, 1200 Brussels, Belgium; 3BTECH, Aarhus University, 7400 Herning, Denmark; 4LuxAI S.A., Luxembourg 1724, Luxembourg; ali.paikan@luxai.com (A.P.); ziafati@luxai.com (P.Z.)

**Keywords:** Internet of Things, socially assistive robots, ambient assisted living, security, process mining

## Abstract

Today’s IoT deployments are highly complex, heterogeneous and constantly changing. This poses severe security challenges such as limited end-to-end security support, lack of cross-platform cross-vertical security interoperability as well as the lack of security services that can be readily applied by security practitioners and third party developers. Overall, these require scalable, decentralized and intelligent IoT security mechanisms and services which are addressed by the SecureIoT project. This paper presents the definition, implementation and validation of a SecureIoT-enabled socially assisted robots (SAR) usage scenario. The aim of the SAR scenario is to integrate and validate the SecureIoT services in the scope of personalized healthcare and ambient assistive living (AAL) scenarios, involving the integration of two AAL platforms, namely QTrobot (QT) and CloudCare2U (CC2U). This includes risk assessment of communications security, predictive analysis of security risks, implementing access control policies to enhance the security of solution, and auditing of the solution against security, safety and privacy guidelines and regulations. Future perspectives include the extension of this security paradigm by securing the integration of healthcare platforms with IoT solutions, such as Healthentia with QTRobot, by means of a system product assurance process for cyber-security in healthcare applications, through the PANACEA toolkit.

## 1. Introduction

In recent years, there has been a host of notorious security incidents across the world. These range from smart security cameras (such as Amazon’s Ring Video Doorbell Pro [1] or Blink XT2 security camera systems [2]) to connected fax machines and printers [3] and even smart light bulbs [4]. The Internet of Things has become an irresistible target for cyber-criminals—thus shaping the threat landscape and the cyber-crime underground [5]. Recent examples include botnets of infected routers and network storage devices [6], the first large scale distributed denial of service (DDoS) attack based on IoT devices [7] as well as vulnerable healthcare devices such as connected pacemakers [8] which, despite reported flaws, have not yet been remediated. These incidents have highlighted the vulnerability of the emerging IoT ecosystem, while at the same time have indicated the high socio-economic importance of IoT security.

Many IoT deployments comprise a plethora of passive or semi-passive devices (e.g., sensors) which might be under the adminsitrative control of a single entity and protected by a security sandbox for their execution. This approach to IoT security is not enough for the wave of emerging IoT applications, which involve intelligent devices with autonomous or semi-autonomous behaviour and smart actuation capabilities as part of larger scale deployments that work across multiple administrative domains. In terms of consumers, healthcare and ambient assisted living platforms can be significantly enriched based on smart objects, such as wearables, socially assistive robots as well as cloud and IoT-based software and hardware platforms, which leads to numerous security issues.

The main contribution of this paper is in detailing IoT security service implementation and lessons learned from the EC-funded SecureIoT project that aims at developing IoT security services in several scenarios. The focus in this paper is on socially assistive robots which aim to provide personalized healthcare and ambient assisted living services. We evaluate several SecureIoT services to provide insights on their feasibility as well as encountered challenges. Thus, the objectives of the paper are:Provide insights related to security challenges in Internet of Things, highlighting the potential use cases and applications for overcoming such challenges, ultimately leading to the SecureIoT project;Comprehensively describe the work underwent for implementing SecureIoT services in scenarios particular for Socially Assistive Robots;Provide an evaluation of SecureIoT services as integrated in SAR scenarios.

The paper is structured as follows: Section 2 presents related studies on IoT layers security, emphasizes the specific threats of socially assistive robots and highlights the framework of SecureIoT project. Section 3 depicts the SecureIoT architecture and the services it provides. Section 4 is dedicated to the implementation of the testbed and it provides the description of the scenarios and of the simulated security threats. In Section 5, the validation methodology and the results are presented while Section 6 presents future perspectives and open challenges. Finally, Section 7 concludes the paper.

## 2. Security in the Internet of Things

### 2.1. Related Work

The Internet of Things has enabled intelligent connection between physical devices and applications in many domains. The Internet of Things have a positive impact on the healthcare domain [9], starting from ambient assisted living applications [10] or applications that target the monitoring of several vital parameters [11] and aiming to build the so-called Internet of Medical Things (IoMT) framework [12]. Even though the IoMT concept is at its early stage, the technologies used to embody each layer of IoMT architectures allow for the collection of health data that may be the premise for new diagnostics and treatments [13]. Nevertheless, when developing IoT-based healthcare applications, one must also assume IoT threats and disadvantages and security measures must be implemented on each layer of the architecture: Devices, Network, Cloud, Application [14] and also on the extensions of the architecture, where applicable, Edge and Fog [14,15], as IoT provides applications such as drug recommendation, sensor calibration, and health parameter monitoring, which may afflict human beings [16]. With the advent of body area network deployment, by transmitting sensitive data over insecure networks [17], the patient privacy and security is endangered, in terms of data confidentiality and integrity [18].

Further, we will highlight the main vulnerabilities of each layer and the methods proposed to improve the security of each component.

The Devices layer comprises lightweight devices with low storage and processing capabilities [19]. Furthermore, the users may not have the appropriate knowledge or skills to assess the malfunctioning or to operate them properly, as in the case of healthcare or ambient assisted living use cases, where non-technical and even cognitive-impaired users are implied [20]. The attacks met on this layer include physical cloning (counterfeiting) [9,21,22], malware, spyware [9] and physical attacks (intentional damage to the device, theft [9] especially for those deployed to work autonomously [23]). Agarwal et al. [24] developed two tools to detect IoT devices in heterogeneous networks and to retrieve device-related data (manufacturer, model, firmware version) and they concluded that many IoT devices manufacturers do not secure the devices at all. When referring to IoT devices, one cannot implement the standard heavyweight, complex techniques [19] to secure the devices belonging to the layer. Nevertheless, there are already two classes of security solutions. On one hand, lightweight, low-power security algorithms were proposed [25]. On the other hand, there is a novel practice to design devices with built-in secure elements [26], following the hardware-based security approach [27]. Another manufacturing-based solution to protect against physical cloning implies the use of physical, unclonable functions [22,28] that assigns the device a unique fingerprint. Another security measure recommends to avoid storing sensitive data on devices [22]. In addition, in order to prevent physical attacks, one should avoid exposing the devices. In the case of ambient assisted living or healthcare systems, this practice can be materialized by also implementing physical security measures.

As the Edge layer is a bridge between IoT devices and the Cloud layer [29], it may be threatened both by physical attacks, jamming attacks, data tampering, intentional data interruptions, eavesdropping and connection flooding [9]. Nevertheless, based on Edge layer existence in IoT architecture, several security designs have been proposed to overtake the challenges imposed by the low-resource IoT devices [29]. The designs are divided into three categories [29]:User-centric design, [30] where on Edge layer a virtual domain in charge of assuring the security of each device is built;Device-centric design [29,31], where Edge layer supports the IoT devices to respond to security requirements;End-to-end design, where Edge layer acts as a secure middleware [29].

The Network layer requires important security mechanisms to assure confidentiality (to make data invisible to the attacker), integrity (to protect data from attacker’s tampering tentatives) and authentication (to allow only verified sources to send messages inside the network) [32]. With respect to the Edge layer, additional cyber-attacks such as man-in-the-middle attacks and relay attacks may occur. Security mechanisms for the healthcare-specific Network layer have also been proposed. One solution resides in using the concept of collaborative networks to detect and to protect against cyber-attacks [33].

The Fog layer represents an interface between IoT devices and the Cloud layer, when the Edge layer is not involved. It is a spatial and processing extension of the Edge layer, that is, the Fog layer is closer to the Cloud layer and has more processing capabilities than the Edge layer. As vulnerabilities, the Fog layer is not resistant to single points of failure, as the Cloud layer is. Furthermore, the Fog layer may afflict data integrity if end-to-end encryption mechanisms are not implemented. Consequently, as in the case of the Edge layer, the Fog layer can bring minor supplementary vulnerabilities, but it is also able to leverage Device layer weaknesses [34]. Thus, architectures involving the Fog layer need to be complemented by security mechanisms that mitigate the vulnerabilities of this layer.

Cloud security in the Internet of Things and Internet of Medical Things was approached in [16] based on a Software Defined Network concept integration in the IoMT architecture. On the other hand, in [35], the patients’ data is protected through a remote lightweight encryption technique based on a Hyperelliptic Curve concept that was implemented on the medical IoT devices before the data was sent to the Cloud layer. Among the attacks that may affect the Cloud layer, one must mention data interruption and intentional outage. Further, another attack exists in the form of a Distributed Denial of Service (DDoS), affecting web servers, storage and other Cloud components such that the Cloud resources can no longer be used. It can be initiated by botnets and the attackers use scanning methods to find vulnerable machines [36]. The prevention techniques include, but are not limited to, ingress/egress filtering [37], route-based distributed packet filtering [38] and secure overlay services [39]. Other attacks include buffer overflow, impersonation and remote code execution [9].

Finally, Application layer security involves the security vulnerabilities and prevention mechanism detected and, respectively, implemented at communication protocol layer. It is worth mentioning IoT protocol solutions such as MQTT, CoAP and less-typical IoT protocols AMQP, DDS, and XMPP [40]. MQTT is widely used in IoT platforms. It is based on Publish/Subscribe messaging and client–broker connections. Further, we will review its vulnerabilities. Firstly, concerning the authentication issues, the broker does not correctly check the identity of publishers or subscribers and even though it experiences repeated authentication attempts, it does not block them, with the following consequences: the attacker gains access to MQTT devices and is able to overload the broker, putting it out of operation. In addition, the MQTT authorization mechanism employed by the broker does not include the correct settings of the publishing/subscribing permissions, with the consequence that the attacker gains control over the data or functions of IoT devices. Moreover, message validation mechanisms that are improperly implemented (for example, a publisher sends messages consisting of invalid characters that cannot be correctly interpreted by brokers and subscribers), facilitate security attacks. Man-in-the-middle (MiTM) attacks [40] may happen as the encryption is performed only for payloads, not for the entire message [41].

To conclude the section, the IoT security is an important subject not only for technical purposes, but also for the welfare of the society and its citizens [40] as, currently, many person-centric applications are being developed.

### 2.2. Socially Assistive Robots

The idea of robotics playing a role in healthcare has been circulated for several years, even decades. The concept of a Socially Assistive Robot (SAR) lies at the intersection of socially interactive robots and assistive robots, an adequate definition being one that gives aid or support to a human user. SARs come in different shapes and sizes, be they humanoid or animal-like (cat, dog, even seal [42]) as well as for different uses such as elderly care, neurodevelopmental disorder improvement, pre-tertiary education [43] and even addressing mental health challenges during a pandemic [44]. However, to date, no scientific literature has reported on security challenges and solutions for Socially Assistive Robots.

It could be argued that a Socially Assistive Robot scenario exposes a significant attack surface. The distribution of the assets and their diversification in terms of applications, operating systems and communication protocols is large. Thus, in the frame of this paper, an indicative (i.e., high level) set of applicable threats will be discussed.

Initially, home-deployed devices and cloud servers constitute distinct targets. The communication channel that interconnects these targets is a target per se since it can be subjected to interception, protocol manipulation, traffic injection & obstruction. Interception can lead to loss of privacy in case sensitive data are extracted or to privilege escalation in case of the interception of administrative accounts.

On the other hand, protocol manipulation and injection can cause the alternation of messages and thus alter the commands of the IoT devices or the server-side business logic. The impact will be to deliver low-quality services causing possible damage to the users/patients, which is considered a breach of contract with financial consequences. Such events will also cause significant damage to the reputation of the product, which is the intangible asset of any company. The same applies in case of privilege escalation, the software stack of the robots can be considered compromised. Finally, any misuse of the programming interfaces or even unintentional misconfiguration of the behavior of the components can lead to issues in the rehabilitation/coaching operations.

The QTrobot is a 63 cm humanoid robot developed by LuxAI for social interaction and teaching applications. The QTrobot has a screen face to show facial emotions and an expressive upper body motion with eight motor controlled degrees of freedom. The QTrobot is powered by an Intel NUC processor, and possesses a 3D camera and a microphone array. QT provides Wi-Fi connectivity as well as a developer-friendly environment using Ubuntu and ROS. The QTrobot is a real Socially Assistive Robot deployed in several assistive scenarios, including rehabilitation and assistance to children with autism. It provides interfaces for programming in C++/Python, along with a high-level language that can be used by carers, therapists and medical experts. It suffers from the same security threats outlined above and needs IoT security services to counteract their effects.

### 2.3. SecureIoT Project

The vision of SecureIoT was to secure the next generation of dynamic, decentralized IoT systems, which span multiple IoT platforms and networks of smart objects, including objects/things with embedded intelligence and (semi)autonomous behaviour. In order to realize this vision, SecureIoT offered a range of open, end-to-end, scalable data-driven security services, which were built around the concept of predictive IoT security. It provides a set of open security services (based on a Security-as-a-Service (SECaaS) paradigm), which can be used by IoT platform providers, IoT solution integrators, and IoT OEMs (Original Equipment Manufacturers) towards securing their products and services in-line with existing and future security regulations. From a technology perspective, SecureIoT offers a set of distributed middleware services and an IoT security knowledge base that supports security audits, risk assessments and programmers’ support for different IoT entities (i.e., IoT devices, smart objects, virtual IoT objects, IoT platforms, cross-platform IoT services) at multiple levels (i.e., Device, Edge/Fog, Cloud).

SecureIoT brought together partners covering the full range of experience and research and development expertise needed in order to realize the project. In particular, the consortium comprised experts in IoT cyber-security, partners with background and experience in data analytics, domain experts with proven solutions in IoT cyber-security as well as legal and regulatory experts.

One of the main characteristics of this approach was the enabling of smart object operation, while at the same time supporting security interoperability in supply chain scenarios that involve multiple IoT systems and platforms with diverse security capabilities. The project, therefore, served as a basis for implementing predictive and intelligent security systems. Figure 1 shows a conceptual architecture of SecureIoT.

From a logical perspective, the SecureIoT platform is structured as a multi-layered security monitoring and enforcement system.

IoT Systems Layer: consists of heterogeneous components such as field devices, fog nodes, field networks, edge gateways and cloud computing infrastructures that make up a typical IoT system. Note that the IoT systems layer is not part of the SecureIoT platform, but rather the layer of field systems that must be secured via the SecureIoT platform and its architecture.Data Collection and Actuation Layer: in charge of interacting with the field (IoT Systems Layer) for collecting security related data from various probes and from all the different parts of IoT systems and driving security-related automation and actuation tasks such as the configuration of security-related properties of IoT systems.Analytics: analyses the collected data (from the Data Collection and Actuation Layer) in order to identify security-related events and indicators in the form of incidents, threats and attacks. It comprises a range of data analytics algorithms (including Machine Learning (ML) and Artificial Intelligence (AI), which are used to detect security events and shape security policies accordingly.IoT Security Services: comprises IoT security services primarily offered by the SecureIoT platform. They are based on the data processing outcomes of the Analytics layer.Use Cases: leverages the security services layer in order to provide security functionalities to specific IoT applications and use cases, such as the Socially Assistive Robot application of the project.

## 3. Security Services in Socially Assistive Robot Scenarios

### 3.1. System Architecture

The SAR usage scenario within SecureIoT provides an extensive testbed to evaluate and validate different components and services of SecureIoT. It consists of a smart object, the QTrobot based on the Linux platform, two cloud platforms—the QTrobot cloud and CC2U cloud—and various modalities of user interaction and data collection including speech communication, and collecting feedback by QTrobot through Android-based tablets, and different sensors such as the QTrobot 3D vision and health and activity sensors of the CC2U smart home. It also includes several types of user roles such as end users, therapists and system admins. Figure 2 summarizes various components of the SAR system and the data flow among them.

The technical integration with SecureIoT is based on data probes which are data collection points installed in different components of the system to send data for security monitoring purposes by SecureIoT. Collection, filtering, aggregation and transmission of data to SecureIoT is performed by integrating the probe system developed by the SecureIoT project. In the SecureIoT architecture, the tool used for central collection and storage of data is Elasticsearch, providing a distributed text based search engine with an HTTP web interface. The suite of Elasticsearch comes with a set of easy to deploy and easy to use tools to collect data, transform them to a desired format and send it to Elasticsearch database:Logstash: is a component for processing and transforming data using filters and send them to Elasticsearch.Kibana: is a web interface for searching and visualizing data.Beats: are lightweight components to collect data from distributed machines and send them to Logstash or Elasticsearch.

The data is then analyzed by the SecureIoT analytics engine to detect abnormalities which could be resulted from various types of attacks such as Man-in-the-middle or Denial of Service attacks. The detected abnormalities are then fed to the Risk Assessment Service developed by SecureIoT for analysis and further suggesting and taking appropriate mitigation actions. In addition to these components, SecureIoT provides programming support and compliance auditing services which are used to implement and audit various privacy and security policies. We will describe the integration, validation setup and results for each of these components and services separately in its dedicated subsections in the rest of this paper.

### 3.2. Data Collection Setup

The data used for the test, development and validation of the SAR scenario is obtained from the following sources, a mix of real interaction data by test users and simulated data:QTrobot Data: sensory data of QTrobot are collected by having test users, internal members of LuxAI, interacting with the QTrobot and using its functionalities such as playing the different games. These data are also stored and can be replayed, simulating the actual recorded interaction, to repeat a scenario several times for test and development purposes.CC2U data: data from the CC2U assisted living environment is simulated using the CC2U simulator. The simulator generates various sensory data related to sleep monitoring, activity monitors and walking steps, for example. A simulated user is driven by models for the home, the weather, the sensing environment and the behaviour of the user in it. The simulator returns all the metadata expected from an actual home, obtained by processing the measurements of the sensors as well as the state of a simulated user, which can be inferred from the measurements.

The SAR system is interfaced to the SecureIoT central database with the followings probes:Generic system probes: collecting the QT’s and the CC2U’s static system configuration as well as their dynamic status. The system data are collected using Metricbeat and Packetbeats, including CPU usage, memory, file system, disk IO, network IO and statistics and statistics of running processes, as well as the data about the network traffic of the system.Application specific probes: collecting application level data. At the component level, a comprehensive logging system has been developed and several probes are installed both at QT and CC2U, providing a fine grain control to the SecureIoT system to start, stop and configure these probes to collect the desired data at a desired rate. The data logged ranges from sensory data such as results of emotion recognition software up to the messages communicated between different components such as coaching messages and game commands exchanged between QTrobot and CC2U. The component’ logs are first collected into log files and are then transferred by Filebeat to Logstash and after processing to Elasticsearch.

An example of the probes in the SAR scenario is the motor state probe which continuously monitors all motors of the QTrobot, logs their position, speed and torque, and sends them to the SecureIoT Elasticsearch database using the SecureIoT data model. Another example is the ROS state probe which continuously monitors the communication state of the components within the QTrobot’s ROS environment, logging the status of publishers and subscribers in ROS and how they are interacting.

### 3.3. Technical Setup

The QTrobots used in the experiments as will be reported in the rest of this section are of two configurations, one QT with the 3D camera and Intel NUC Processor with Ubuntu 16.04 to implement the second game of SAR-1 scenario, and another QT with a 2D camera and Raspberry Pi, which is a processor with an Arm architecture. The latter architecture is widely used in majority of robots in the market and was chosen to make this setup closer to an actual setup of IoT applications with limited processing and memory resources. Both setups run the same ROS software and API and are seamlessly integrated with CC2U. However, the second setup imposes harder requirements on probes and SECaaS regarding consumed system resources, providing a great testbed for their evaluation in constrained setups.

### 3.4. Technical Evaluation Methodology

To validate the SecureIoT services an end to end demonstration has been set up utilizing most of the SecureIoT infrastructure. More specifically, the following main components have been used, (see Section 2.3):SecureIoT Probes/Data collection layer: for pushing collected raw measurements to SecureIoT Infrastructure.Data Routing/Analytics layer: for storing the data pushed from the probes to the Global Repository (ElasticSearch),Security Template Extraction/Analytics layer: for training the Analytics algorithms with annotated historical data coming from the IoT Systems,Analytics Engine/Analytics layer: which is using the trained templates to analyze the real time data coming from the IoT Systems and are stored to the SecureIoT Global Repository (ElasticSearch),Data Bus: which is used as a messaging channel implementing a publish/subscribe paradigm for the Analytics reports,CMDB/IoT Security Services layer: where specific use case data describing the involved assets and the potential vulnerabilities/threats are stored,Risk Assessment Engine/IoT Security Services layer: which is analyzing the published reports from the Analytics Engine to the Data Bus andRisk Assessment Dashboard/IoT Security Services layer: which is responsible to visualize the risk assessment reports with possible mitigation actions.

Note that, for the sake of brevity, these components have not been explicitly mentioned in the architecture in Figure 1, but are referred in the list together with the layer in the architecture that they belong to.

For the SAR validation, we are focussing on the directly related components, which are the CMDB, Analytics Engine, Data Bus, Risk Assessment Engine and Risk Assessment Dashboard. The other components are not being directly used by our SAR scenario, but rather indirectly through other components (e.g., Global Repository). Figure 3 depicts a high-level dataflow of the validation runtime.

More specifically, we can see that, first, we instantiate the Risk Assessment engine which retrieves all the configuration data from the CMDB to be initialized (not depicted in the sequence diagram) and starts listening for the Analytics Engine reports by subscribing to a Data Bus topic named after the ID of the Analytics Engine. Then, the Analytics Engine consumes incoming data from the ElasticSearch (Global Repository) where the captured Raw Data are stored. The data are then analyzed for specific attacks. When an attack is identified, it is pushed to the Data Bus by using as the topic name the ID of the instantiated Analytics Algorithm. The report is then picked up from the Risk Assessment Engine which applies preconfigured rules for the specific scenario and produces a report with possible mitigation actions. This report is then picked up from the Risk Assessment Dashboard in order to provide it to the end user.

### 3.5. Methodology for Stakeholder Feedback

Within the SecureIoT project, the consortium proposed both a qualitative as well as a quantitative analysis of stakeholder feedback, based on three pillars:A set of interviews conducted with a few stakeholders close to each use case (with questions in free form).Filling in a stakeholder questionnaire with a large pool of stakeholders thus providing as many meaningful results as possible.Filling in a User Experience Questionnaire (UEQ) that automatically calculates feedback results and outputs meaningful graphs. The motivation for using the UEQ is that at the final stage of the SecureIoT project, User Experience is warranted to be validated. The UEQ relies on a well-known standard available online.

The stakeholder feedback was conducted, but for this paper we focus only on the presentation of the feedback for Socially Assistive Robots scenarios. Since the first two pillars are difficult to be quantified and are out of scope for this paper, we present results from the User Experience Questionnaire.

Within the UEQ evaluation conducted for the project, we considered the short version of the UEQ. This short version of the UEQ consists of just eight items. Four of these items represent pragmatic quality (items such as obstructive/supportive, inefficient/efficient, complicated/easy and confusing/clear of the full UEQ) and four hedonic quality aspects (items such as boring/exciting, not interesting/interesting, conventional/inventive, usual/leading edge of the full UEQ). Thus, the short version does not provide a measurement on all UEQ scales. It consists of the scales *Pragmatic Quality* and *Hedonic Quality*. In addition, an overall scale (mean of all eight items) is reported.

## 4. Implementation

### 4.1. Scenario Description

The aim of the SAR usage scenario is to evaluate and validate the SecureIoT services in the scope of personalized healthcare and ambient assistive living scenarios, representing a rich set of hardware and software assets and types of data. The SecureIoT SAR usage scenario demonstrates the secure integration of LuxAI’s QTrobots in the CloudCare2U (CC2U), an IoT-enabled healthcare platform of iSprint’s, to enable the delivery of personalized ambient assisted living functionalities such as rehabilitation and coaching exercises by QTrobot within a wider rehabilitation and coaching programs offered by CC2U.

#### 4.1.1. Scenario SAR 1—Cognitive and Physical Games

In the SAR-1, QTrobot (QT) offers two types of interactive and engaging games designed to stimulate the memory functioning and physical activities of the elderly. These include a card game in which the QT and the user have to remember the sequence of cards they are showing to each other, and a similar memory game where instead of showing cards, QT and the user should remember and replicate various hand gestures such as right-hand-raised-up and left-hand-opened-to-the-left that they are showing to each other. In the latter case, the user exercises more physical activity while playing the memory game.

As part of this scenario, a bidirectional integration of the CC2U cloud platform and QTrobot (QT) was realized where coaching activities such as the cognitive and physical games described above are initiated by CC2U, played between QT and user and the results are sent back, stored and visualized in CC2U cloud platform and its eWALL interface, shown in Figure 4. In addition, several data probes were installed, as will be detailed in the rest of this section, to collect and send various types of data to the SecureIoT server.

#### 4.1.2. Scenario SAR 2—Monitoring and Check-Ups

This scenario extends the SAR-1 by introducing more types of platforms and data to the system. A tablet interaction is introduced with which the end user can answer to the health related questionnaires asked by QTrobot. Results are then sent back the cloud. In addition, various other data such as user emotion and posture from QTrobot (Figure 5), as well sensory data from CC2U gateway such as vital signs and sleep measures are collected and sent back to the CC2U and QTrobot cloud for visualization, reporting and monitoring. Related probes are also installed to stream these new data to SecureIoT for security monitoring.

#### 4.1.3. Scenario SAR 3—Daily Calendar and System Admin

SAR 3 extends the previous scenarios with panels and corresponding access control functionalities for the end users and their therapists to access, visualize and monitor activities and health data such as results of the games and health questionnaires. Furthermore, it provides a calendar functionality as shown in Figure 6 where end users can set reminders, and professionals (e.g., therapists) can schedule different activities such as QTrobot games or questionnaires for their associated patients (end users), implemented along with corresponding access control policies.

### 4.2. Predictive Analytics and Risk Assessment

The predictive analytics framework of SecureIoT was validated by providing various data sets each containing data recorded from a normal state of the system as well as data representing abnormal cases. The following types of data were recorded from the actual QTrobot, and the CC2U simulator producing data of a user behavior in an ambient assisted living home.

Generic system data: such as data related to CPU, memory, disk and network statistics;Low level system data: such as patterns of motors data, patterns of messages passed between ROS components and the frequency of exchanges, and patterns of communication messages between CC2U components;Application-level data: history of played games and their results as well as patterns of gestures used in the games and patterns of vital sign data and number of steps.

The abnormal cases represented two types of attacks, man-in-the-middle and denial of service.

Man-in-the-middle attack: data for abnormal cases were created to represent attacks tampering the system behavior and/or its data Figure 7. For instance, the positions of motors in a normal state of the system follow a specific profile, hence creating a regular data pattern. To create abnormal data sets, new gestures were introduced and played on the robot representing the alteration of its normal behavior, creating abnormal motors position and speed, and hence generating irregular patterns. The abnormal data sets were annotated with attack times to enable development and testing of data analytics algorithms.

DoS attack: for simulating a DDoS attack, we have used a variant of the HULK DDoS attack tool. HULK is a DDoS attack similar to an HTTP flood which is designed to overwhelm web servers’ resources by continuously requesting single or multiple URL’s from many source attacking machines. The HULK flood differs from most available DDoS attack tools which produced predictable repeated patterns that could easily be mitigated. The principle behind the HULK flood is that a unique pattern is generated at each request, with the intention of increasing the load on the servers as well as evading any intrusion detection and prevention systems.

While there are other types of attacks in IoT scenarios, these two were the focus of the validation activities within the SecureIoT project. Other types of attacks were used by other use cases within the project.

Based on several data sets provided, the analytics algorithms were developed and tested by project partners to report the corresponding metrics.

For the Risk Assessment, the methodology used to model the threat scenarios consisted of the following steps:Capturing information from the expert of the scenario;Refining the cyber-threat scenarios;Modelling the cyber-threat scenarios and building the risk assessment model in the system including creating a mathematic model about how the system behaves, relations between elements and probabilities of threats to happen.

The risk assessment of the Socially Assistive Robot usage scenario was performed for three attack scenarios: man-in-the-middle, DoS and malware (ransomware) attacks. This was performed iteratively in order to increase the number of attack scenarios and data so we could test with more information the risk assessment solution. Although the assets remain similar in the scenarios the threats, attacks, indicators, recommendations, impact, etc., are different so we could have much more data for the risk assessment solution. Following we present a representation of the malware attack of the scenario in Figure 6, where we can see represented the different assets, threats, impact, indicators, etc. This information was then used as basis for creating the information that is used in the risk assessment solution.

The model represents the three assets that could be infected with malware, indicators and impact in the system. This model represents the scenario but does not show all the data that we later use in the solution for detecting and informing about the attack. As an example, in Figure 8 and Figure 9 we show how the information of the models is transformed into machine-oriented data and used for the calculation of risks (and support for the user) in the risk assessment solution. The user would receive information about how to react to the threats also according to the level of risk detected, so we also differentiate different levels of risk and, therefore, different reactions to them.

## 5. Evaluation

### 5.1. Data Collection and Probe Validation

After multiple measurements on both configurations of QTrobot (QTrobot-TP with Raspberry 3+ model B, and QTrobot-RD with NUC i5 Processor), our observation is that the overhead of probes in terms of CPU and memory overhead is negligible. Both QTrobot’s configurations in this case are quite decent in terms of computational capability and while the CPU and memory overhead might be potentially a concern in IoT devices with very limited processing power, it is not the case for our Socially Assistive Robot use case. Regarding the network overhead, QTrobot is connected to Internet through a standard WiFi interface and the network overhead observed does not pose any issue for our use case. However, the network overhead may become an issue in settings with limited network bandwidth. Table 1 shows the measurements taken of the network load introduced by different probes. The value represented in the table can be converted to network throughput (Bit/s), but to show their relation to the publishing rate, we present them in term of KB overhead per each publish.

As can be seen Table 1, the highest overload is related to the motor state probe (i.e., “sar_motorstate”) which continuously monitors and publishes high frequency motors’ state data. However, this probe can be configured to log data with a lower rate or only when it is required by the Data Analytic engine. An example of such situation is the scenario in which, as soon as a pre-attack is detected by data analytic engine, the motor state probes will be configured to provide more data for the engine to detect the potential abnormality and eventually any attack.

### 5.2. Predictive Analytics and Risk Assessment

#### 5.2.1. Anomaly Detection

To detect anomalies in SAR data, process mining was used. Extensive series of experiments were performed in order to evaluate and compare the performance of process mining to alternative commonly used detection methods, including elliptic envelope, support-vector machine, local outlier factor and isolation forest techniques, also considering the influence of the time splitting, the influence of the pre-processing clustering and the detection performance according to multiple criteria. A more thorough description of the process mining methodology used can be found in our previous work [45]. Further consideration is also given next.

The data involves measurements for all the available WiFi networks at any time, but the only important one is the one that the QTrobot connects to (in this case, the one with the “QT Demo” SSID). Still, a single input of collected information can contain many measurements attributed to the SSID in question, but a variational autoencoder can accept inputs of a fixed, predefined shape. To this end, we statistically combined all samples attributed to the same SSID from a single input, forwarding the sample mean and variance to the model. The platform collects features such as the frequency and the signal strength percentage; these values are inherently bounded, and we do not need to force an explicit upper down (but we still downscale for numerical stability).

The training and validation sets are isolated on a sequence basis, as the first two sequences are used for training and the third for validation. The “attack” inputs are once more biased with a higher-class weight to account for their lower number.

Using this training process, the convergent model produced the validation metrics. Figure 10 presents the Precision versus Recall for the man-in-the-middle attack dataset. The classification threshold is the normal class reconstruction loss. The choice of the loss threshold above which (pre-)attacks are reported affects the performance presented in Figure 11 which shows the general statistics (accuracy, false positive rate, false negative rate) vs. the reconstruction error threshold.

This “segmented” view of the graphs is expected, as the compressive properties of autoencoders can cause discretisation. Using a threshold of 0.012 on the field provides optimal performance. Compared to usual scenarios, the SAR scenario demanded a more complex model and training process. This case involves analogue measurements, which naturally contain inherent variance (noise). As is evident, variational autoencoders provide denoising properties that are useful in similar cases.

#### 5.2.2. Validation of Predictive Analytics and Risk Assessment Service

In order to validate the predictive analytics and risk assessment service, a testbed was developed where attacks are simulated, data is processed by the predictive analysis algorithms to detect the anomalies, and results are then fed to the risk assessment engine to analyze the risks and take appropriate mitigation actions.

We implemented an attack scenario to imitate a man-in-the-middle and data tampering attack where the attacker forces the robot to route all its network traffic to the attacker router. This is carried out by introducing a stronger (higher signal strength) access point with similar SSID and credentials as the one the robot is connected to. The attacker then can access the robot network, sniff, tamper or inject its own data (i.e., abnormal robot gesture). The network data, status of robot Wi-Fi connections along with components configuration (connections) information are monitored by corresponding probes and sent to the SecureIoT central database. The risk assessment engine is then identified by the analytics engine when an attack is recognized by it.

Assets setup:Two access points with the same SSID, one for normal connection and the other for attack imitation;Three probes:
-“sar_wlan”: to monitor and collect available access points information along with the active one (the one that robot is connected to);-“sar_rosgraph”: to monitor and collect data about the ROS internal state (publish/subscribe components);-“sar_motorstate”: to monitor and collect all motors’ positions, velocity and efforts.User, who is playing with the robot;Attack generator (i.e., “attack_gesture”) which disturbs robot normal behavior by playing unrelated gesture on the robot;Attacker who turns on the second access point, access the robot ROS network and run the attack generator.

The normal situation consists of the following:Only one of the two access points is on;User is playing a game (memory game) with the robot;Robot plays only specific gestures so the motors follow specific position trajectory.

The attack Steps are the following:Attacker turns on the second access point with the same SSID, in which case, the “sar_wlan” probe reports the second access point;Attacker runs the attack generator to disturb robot normal behavior, in which case, the “sar_rosgraph” probe reports a new software component (i.e., “attack_gesture”) in the ROS network;Robot plays gestures, which are unrelated in the current application context, in which case the data reported by “sar_motorstate” contains abnormal trajectories of motors’ positions.

#### 5.2.3. Predictive Data Analysis

A specific gesture was played repeatedly on the robot with some intervals. At the same time, an offline probe logs all robot’s motor positions, velocity and torque. In normal cases, motors position follows a specific profile, hence creating regular data pattern. At some random intervals, the abnormal cases were generated by playing some random gestures different from the actual one or with different speeds. These abnormal motors position and speed create irregular patterns. The abnormal cases (irregular patterns) are annotated in the provided data as index against attack detection feasibility check by predictive analysis algorithm. Figure 12 demonstrates the result of abnormal cases detection by data prediction and analytics engine of SecureIoT on QTRobot motor-state dataset. As shown in the figure, the green line indicates the actual abnormal cases which occurred over the time and the red line indicates successful detection by analytics engine. A more detailed analysis of SAR data can be found in [46].

### 5.3. Stakeholder Feedback Evaluation

The UEQ was conducted with 18 participants, where the objective of this questionnaire was to evaluate the level of quality of the user experience of the SecureIoT prototype. The participants were provided with the multiple items as described in Section 3.5. From the data of the 18 participants, the results can be interpreted that values between −0.8 and +0.8 represent a neutral evaluation. Values greater than 0.8 is quite positive, and less than −0.8 is negative. Table 2 shows the mean, variance and standard deviation of the UEQ scales input by the participants.

In order to see what these figures represent, we show a benchmark against existing datasets which can be seen in Figure 13.

To get a better picture on the quality of a product it is necessary to compare the measured user experience of the product to results of other established products, for example from a benchmark data set containing quite different typical products. It contains at the time of writing the data of 452 product evaluations with the UEQ (with a total of 20190 participants in all evaluations).

The benchmark classifies a product into five categories (per scale):Excellent: In the range of the 10% best results;Good: 10% of the results are better and 75% of the results are worse;Above average: 25% of the results in the benchmark are better, 50% of the results are worse;Below average: 50% of the results in the benchmark are better, 25% of the results are worse;Bad: In the range of the 25% worst results.

We can see that it represents an overall good score against existing UEQ studies.

## 6. Future Perspectives and Open Challenges

The SecureIoT project enabled the integration and application of the developed IoT security solutions/components into the three completely diverse and complex demonstrators. Within this paper we have focused on the Socially Assistive Robots use case. During the project duration, we identified and described the abuse cases for the SAR IoT domain. Based on a threat assessment methodology we explored the used assets, the type of communication between these assets and their usage patterns. Then, we proceeded by identifying relevant vulnerabilities and corresponding threats (CAPEC). The exploration of all scenarios and abuse cases led to an exhaustive listing of attack patterns which we analyzed and managed to identify how threat management can be improved in the SAR scenarios with the SecureIoT SECaaS.

The stakeholders favored the overall technological concept of SecureIoT and the practicalities of SecureIoT’s ambition to continuously track mainstream (and custom) vulnerabilities and threats. They also enjoyed the level of monitoring and control they have over the assets, something that is quite attractive for a deployment of hundreds of thousands of systems.

What the experience from the SecureIoT project has shown is that there is the need of highly programmable security services that can be easily deployed and integrated in existing IoT ecosystems or can be used for building new IoT platform offerings. The entire combination of SecureIoT architecture, data, analysis, services and validation (both technical as well as economical and stakeholder feedback) suggests the migration towards a *real-life security toolbox* offering.

This, in turn, brings forwards new business models for security-as-a-service for IoT services spanning multiple platforms and ecosystems, enabling the functioning of marketplaces for such services. Such a marketplace would be a platform of threat models and IoT security policies for different use cases in various application domains, which would enable its members to create business services as well as set pricing and charge independently for their offerings.

The IoT security services brought forward by this work can serve to secure integrations and federations of furhter healthcare platforms. One example is the integration of Healthentia [47] with QTRobot, by means of a product assurance process for cyber-security in healthcare applications, following the PANACEA paradigm [48]. PANACEA is a people-centric cybersecurity solution for healthcare that offers toolkits for cyber security assessment and preparedness of healthcare ICT infrastructures and connected devices, namely the Solution Toolkit and the Delivery Toolkit. PANACEA supports dynamic risk assessment & mitigation (threat modelling, attack modelling, response management, visual analytics), blockchain for secure information sharing, identification/authentication (cryptographic authentication protocols, biometric recognition/digital identity, IoMT identification),secure behaviour decision models and influences.

In a specific example of the integration of Healthentia with QTrobot and the use of the PANACEA toolkit, Healthentia communicates with QTrobot to facilitate interaction with patients and enhance data collection. Certain patient categories (e.g., young children, older adults) consider interacting with an expressive robot like QTrobot more natural than tapping on touchscreens. Such patients will be answering questionnaires forwarded to them by Healthentia via the QTrobot. During their interaction, the QTrobot also measures their body posture and facial expression, and reports both back to Healthentia along with the answers the patients have provided.

In this specific example the interaction between the two may be protected by enabling the SECaaS programming support services.The SECaaS services would enable the programmatic enabling and disabling of access control policies needed to access the Healthentia platform.

## 7. Conclusions

The papers addressed the security threats of each layer of the Internet of Things architecture, in general, and of an IoT-based healthcare system, in particular, with an focus on Socially Assistive Robots that may be integrated in IoT environments in personalized healthcare and Ambient Assisted Living scenarios.

The SAR scenarios were integrated into a SecureIoT architecture aimed at validating the services via an end-to-end demonstrator setup to showcase most of the existing SecureIoT infrastructure. The validation was performed both from a technical perspective, but also as stakeholder validation for the future marketization of the project services.

The experience shows that a security toolbox offering for IoT scenarios, especially for healthcare services, is critical, both for ensuring the security of such a platform and easing the organisation workload, but also for enabling new business models.

## Figures and Tables

**Figure 1 sensors-21-06912-f001:**
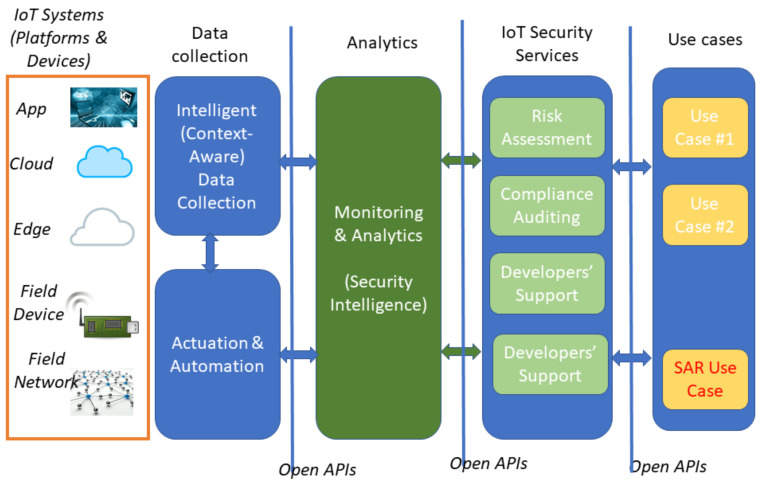
Secure IoT Architecture overview.

**Figure 2 sensors-21-06912-f002:**
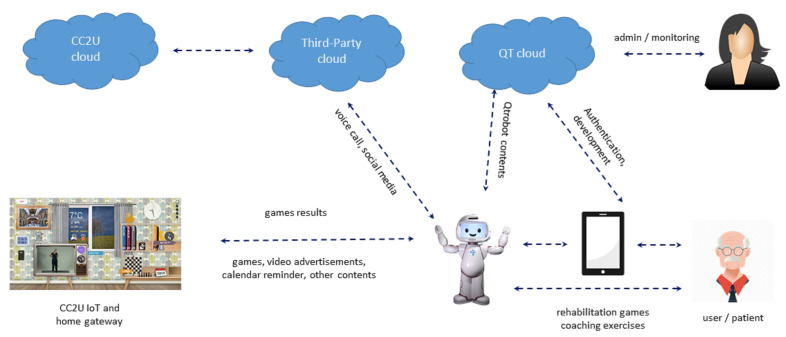
Data flow in SAR scenarios.

**Figure 3 sensors-21-06912-f003:**
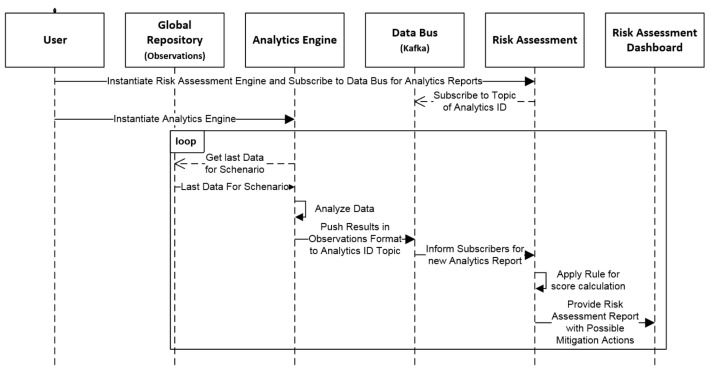
Data Management/Analytics/Risk Assessment Validation High Level Sequence Diagram.

**Figure 4 sensors-21-06912-f004:**
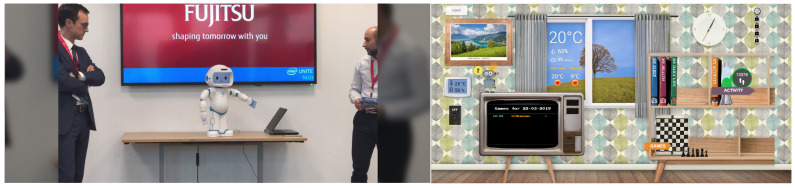
QTrobot playing games and results are uploaded and shown in the eWALL interface of CC2U platform.

**Figure 5 sensors-21-06912-f005:**
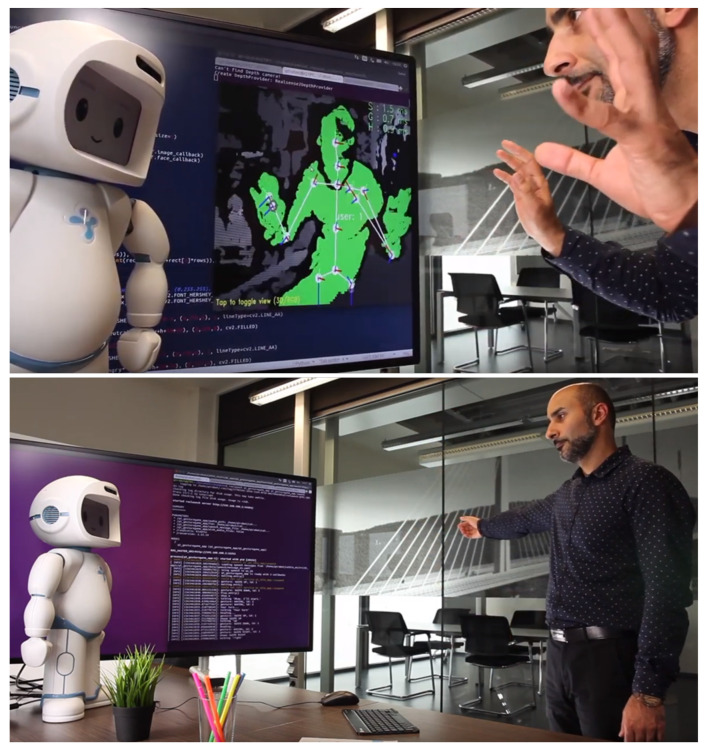
QTrobot collecting facial emotions and body postures of user during interaction.

**Figure 6 sensors-21-06912-f006:**
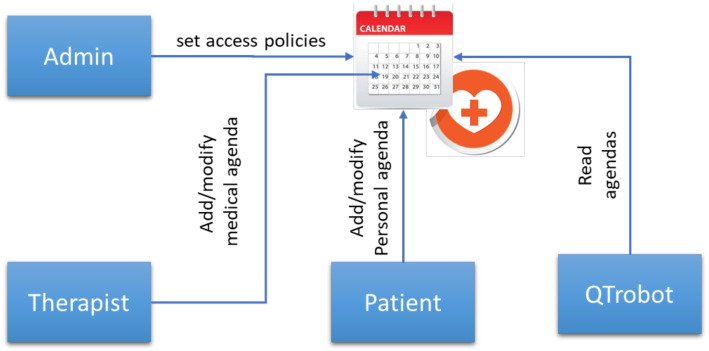
Simplified representation of medical calendar access control.

**Figure 7 sensors-21-06912-f007:**
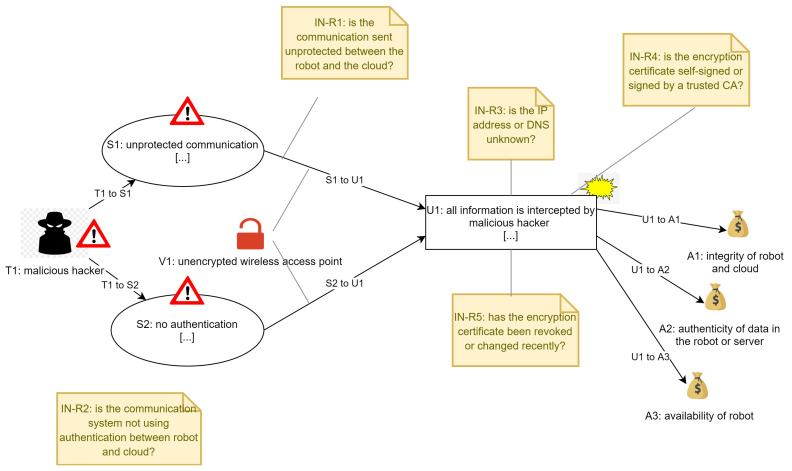
Representation of the model for man-in-the-middle attack.

**Figure 8 sensors-21-06912-f008:**
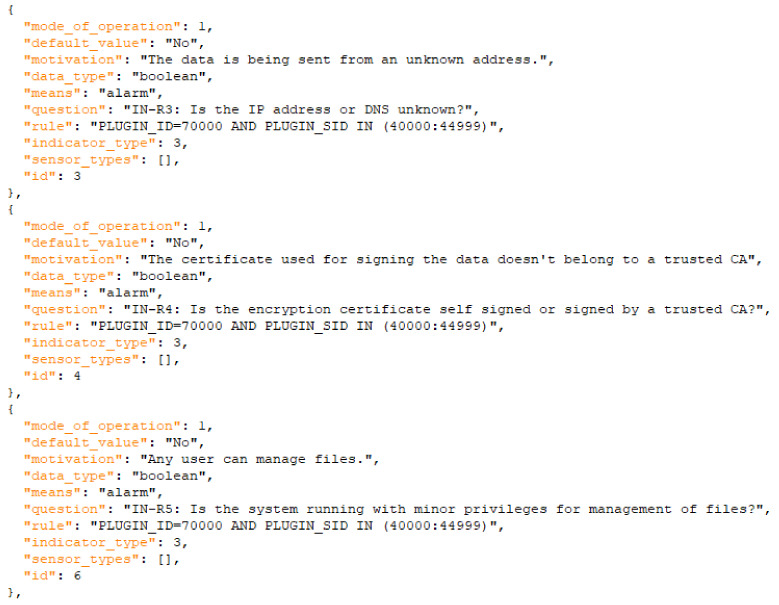
Extract of indicators of the DoS scenario.

**Figure 9 sensors-21-06912-f009:**
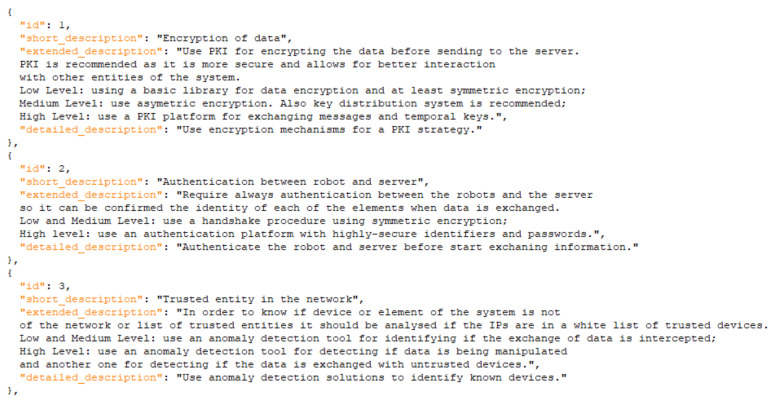
Extract of mitigations of the DoS scenario.

**Figure 10 sensors-21-06912-f010:**
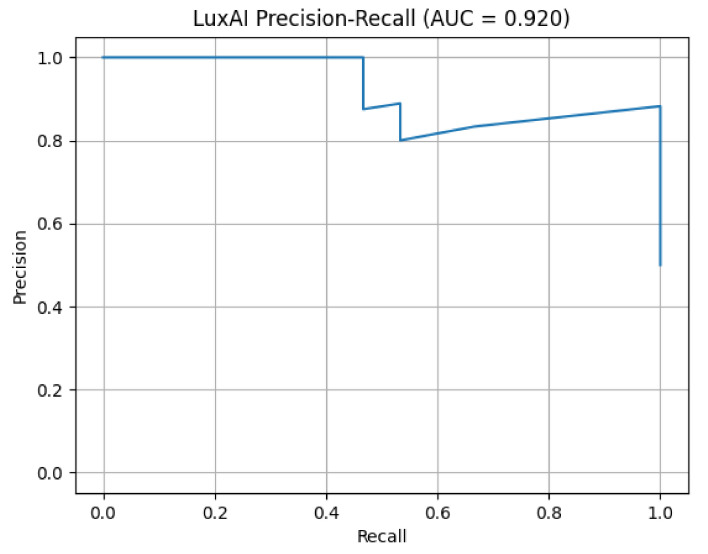
Precision vs. recall for man-in-the-middle attack.

**Figure 11 sensors-21-06912-f011:**
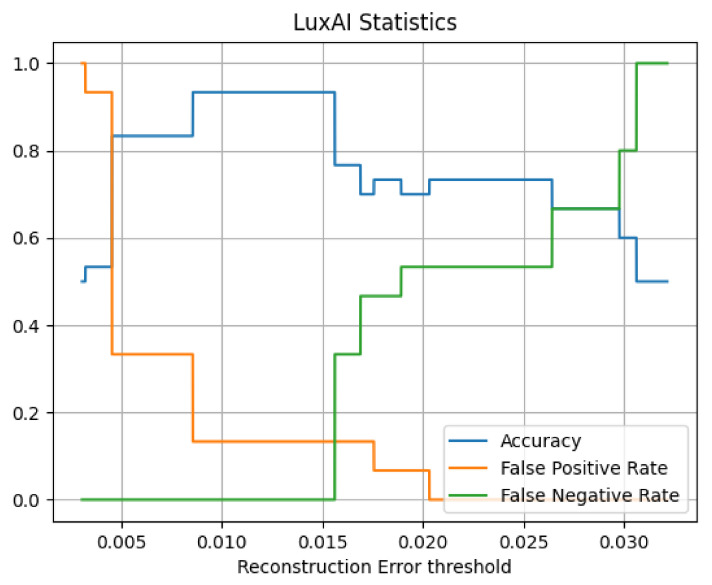
Man-in-the-middle attack statistics.

**Figure 12 sensors-21-06912-f012:**
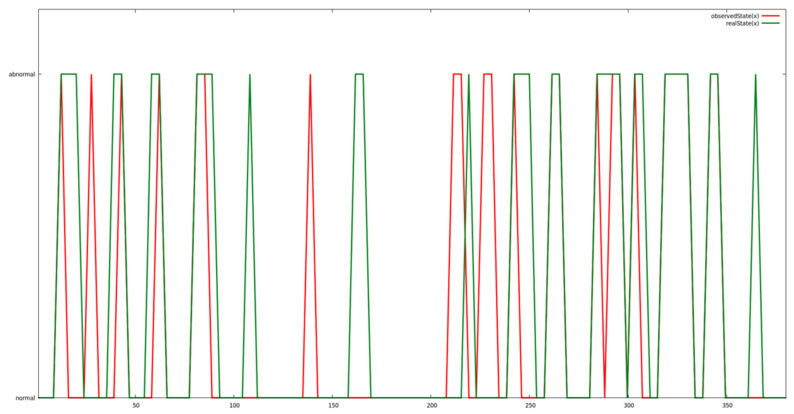
Results of abnormal cases detection by data prediction and analytic engine on QTRobot motor-state dataset.

**Figure 13 sensors-21-06912-f013:**
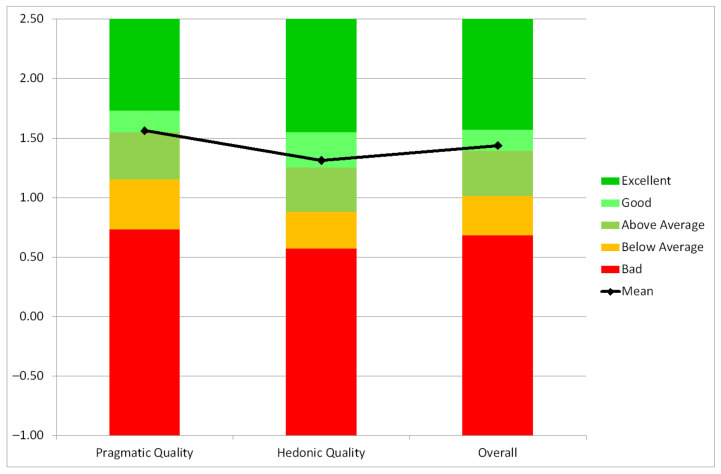
UEQ Benchmark in SAR scenarios.

**Table 1 sensors-21-06912-t001:** Network load introduced by probes.

Probe ID	Sampling Rate	Publish Rate	Sample Size	Network Overhead (per Each Publish)
sar_motorstate: QTPC	5 Hz	0.1 Hz	1.4 KB	70 KB
sar_wlan: QTP	0.1 Hz	0.1 Hz	0.8 K	0.8 K
sar_rosgraph: QTPC	0.1 Hz	0.1 Hz	6.5 K	6.5 K
Total network overhead per each publish (10 s):				77.3 K

**Table 2 sensors-21-06912-t002:** UEQ short scale values for SAR scenarios.

Item	Mean	Variance	Std. Dev.	No.	Negative	Positive	Scale
1	1.7	0.8	0.9	18	obstructive	supportive	Pragmatic Quality
2	1.2	1.0	1.0	18	complicated	easy	Pragmatic Quality
3	1.7	1.2	1.1	18	inefficient	efficient	Pragmatic Quality
4	1.7	1.3	1.1	18	confusing	clear	Pragmatic Quality
5	1.3	0.9	1.0	18	boring	exciting	Hedonic Quality
6	1.6	1.2	1.1	18	not interesting	interesting	Hedonic Quality
7	1.2	1.0	1.0	18	conventional	inventive	Hedonic Quality
8	1.1	0.6	0.8	18	usual	leading edge	Hedonic Quality

## Data Availability

The data presented in this study are available on request from the corresponding author.

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
