# Peer review of "Enabling Security Services in Socially Assistive Robot Scenarios for Healthcare Applications"

_sensors, 2021, doi:10.3390/s21206912_

Round 1

Reviewer 1 Report

This study has presented something is really practical and interesting.

  1. 2. Fig. 8, Fig. 13 the words are small and not clear, please modify it.
  2. This study is not easy reading, it is difficult to find the proposed scheme of the author. Although the quality of the simulation is good.
  3. Could please provides more details about Fig.14, why the sizes of different zones are different?
  4. Why did you analyze two kinds of attacks? There are many attacks in the IoT scenario.

Reviewer 2 Report

The authors of this paper present the definition, implementation and validation of a SecureIoT-enabled socially assisted robots (SAR) usage scenario. According to the authors (which is well presented and validated in their paper) the aim of the SAR scenario is to integrate and validate the SecureIoT services in the scope of personalized healthcare and ambient assistive living (AAL) scenarios, involving the integration of AAL platforms, namely QTrobot and CloudCare2U.

This is a very well written and interesting paper. The authors have clearly presented their work which surely deserves publication. Some minor typos and/or syntax errors exist in the paper (see e.g., lines 481, 617, 619) can be corrected during the preparation of the camera ready version.

Reviewer 3 Report

Although the subject of the work could be of interest in the field of security methodologies and techniques in distributed environments based on IoT, the document presents a series of deficiencies that discourage its publication in Sensors magazine.

Summary of the review of the main aspects of the document:

Organization and Style: The document is well organized but difficult to read for a reader interested in the usual contents of Sensors magazine.
Technical Accuracy: The paper seems technically accurate, but it's not clear if the contribution made is relevant. 
Presentation: The presentation of the work is formally correct but has some lack os rigour from a methodological point of view.
Adequacy of Citations: adequate but incomplete

The main drawback is that it is not clear if the contribution of the work is relevant for the readers of Sensors Mag.

The content is highly specialized and its understanding requires a level of prior knowledge on the part of the reader that greatly limits its interest to an average reader. As a most notable example we have figures 1 and 2, that are tremendously complex, full of details and incomprehensible to anyone who is not a specialist in the subject of the work. 

On the other hand, a framework of security services in environments that use socially assistive robots is proposed, but at no time is the originality or validity of the proposal presented justified. Specifically, point 2.2 (Socially Assistive Robots) of section two (Security in Internet-of-Things) does not provide any reference to other similar proposals or with which some type of comparison can be established. In this sense, it cannot be determined whether the proposal presented is relevant or whether there are more suitable alternatives to solve the problem posed. 

Finally, I consider that the theme of the work does not fit into the editorial line of the magazine and therefore lacks interest for regular readers. In this sense, my recommendation to the authors is to send this work to another journal with an editorial line that is more appropriate to the topic they address. 

Round 2

Reviewer 1 Report

The paper is revised and I feel is ready for publication.

Reviewer 3 Report

The authors have taken into account the suggestions made in a satisfactory way. 

My opinion remains that, even in the context of a monographic issue such as this one, the content of the paper is of little interest to the regular reader of Sensors Magazine, but this is an editorial decision.